# Trends and disparities in antenatal care utilization: A multilevel analysis of four-plus and eight-plus ANC visits among urban women in Ethiopia

Worku Dechassa Heyi[1]*, Wubegzier Mekonnen Ayele[1], Dessalegn Y. Melesse[2]

**1** Department of Reproductive, Family and Population Health, School of Public Health, College of Health Sciences, Addis Ababa University, Ethiopia, **2** Institute for Global Public Health, University of Manitoba, Winnipeg, Canada

* worku.dechassa@aau.edu.et

## Abstract

### Introduction

Antenatal care(ANC) is the gateway of the continuum of reproductive health care, offering a framework for complication prevention, screening, diagnosis, and health promotion.

### Objective

This study examined trends, disparities and associated factors of ANC utilization (ANC4+ and ANC8+) among urban women in Ethiopia between 2000 and 2019.

### Methods

We analyzed five rounds of Ethiopian Demographic and Health Surveys (2000–2019) data, disaggregated by wealth, education, and region using Stata version 16.1. Multilevel logistic regression model was used to identify factors associated with four-plus and eight-plus ANC utilization. Adjusted odds ratios (AORs) with 95% confidence intervals were reported, with statistical significance set at $p < 0.05$.

### Results

ANC4+ utilization increased from 43.1% in 2000 to 58.7% in 2019. while, ANC8+ utilization declined from 11.4% to 3.6%. Among the urban non-poor, ANC4+ rose from 60.5% to 81.8% compared to 37.8% to 48.7% among the urban poor. Regional disparities were pronounced: Addis Ababa showed the highest ANC4+ use (72% to 82%), but a decline in ANC8+ (38% to 10%). Somali region had the lowest ANC4+ (9% to 24%) utilization, though it increased from 9% in 2000 to 24% in 2019 and Gambella had the lowest ANC8+ utilization. Wealth disparities were strong, with

**Data availability statement:** All relevant data are within the manuscript and its Supporting Information files.

**Funding:** The author(s) received no specific funding for this work.

**Competing interests:** The authors have declared that no competing interests exist.

**Abbreviations:** ANC, Antenatal Care; AOR, Adjusted Odds Ratio; EDHS, Ethiopian Demographic Health Survey; FMoH, Federal Ministry of Health; HSDP, Health Sector Development Program; SDG, Sustainable Development Goal; SPH, School of Public Health; SSA, Sub-Saharan Africa; UHC, Universal Health Coverage; VIF, Variance Inflation Factors; WHO, World Health Organization.

non-poor women more likely to utilize ANC4+ (AOR= 2.43; 95% CI: 2.12–2.78) and ANC8+ (AOR= 3.02; 95% CI: 2.49–3.65) than poor women. Education also significantly associated: women with secondary or higher education had greater odds of ANC4+ (AOR= 3.02; 95% CI: 2.54–3.59) and ANC8+ (AOR= 2.13; 95% CI: 1.63–2.78). Women aged 30–34 years had higher odds of utilizing ANC4+ (AOR= 2.89; 95% CI: 2.10–3.98) and ANC8+ (AOR= 2.22; 95% CI: 1.37–3.59). Weekly media exposure was positively associated with ANC4+ (AOR= 1.53; 95% CI: 1.28–1.83).

## Conclusion

Significant disparities in ANC (ANC4+ and ANC8+) utilization persist across wealth, education and regions in urban Ethiopia. Wealth status was the strongest predictor, with non-poor women steadily advantaged. Maternal education, age, media exposure, and marital status also showed positive associations, while regional disparities persisted, with Addis Ababa outperforming other urban areas. Improving ANC uptake requires targeted interventions for urban poor women, alongside media-based awareness and educational initiatives.

## Introduction

According to the World Health Organization (WHO), maternal health refers to the well-being of women throughout pregnancy, childbirth, and the postpartum period [1]. Maternal health services are crucial, as they can save millions of lives among women of reproductive age [2]. However, about 800 women worldwide die daily from pregnancy or childbirth complications, with an estimated 287,00 deaths occurring in 2021 [3]. Ninety-nine percent of maternal deaths take place in developing nations, while more than half of these fatalities take place in sub-Saharan Africa (SSA). Ethiopia is responsible for nearly 3.6% of maternal deaths in the world [4]. The high rate of maternal deaths is due, in part, to unequal access to services for maternal health, and it emphasizes the wealth gap [2].

The under-utilization of maternal health service, particularly in developing countries, is responsible for the high rate of maternal mortality. Maternal health services utilization refers to how often pregnant women use antenatal care(the gateway for reproductive health services, and it is a vital component of the reproductive healthcare continuum that offers a framework for critical healthcare functions such as complication prevention, screening, diagnosis, and health promotion), delivery, and postnatal care services [5].

Despite significant investments in maternal health services over the past 25 years, including prioritization within the national reproductive health strategy, utilization remains low in Ethiopia. This gap between service availability and utilization contributes to the country's persistent maternal health challenges [6]. On the other hand, despite their demonstrated value in reducing obstetric risks [7,8], not all urban residents utilize ANC services [9]. In line with this, it can be more difficult for low-income urban populations, especially those residing in slum areas and squatter settlements, to access quality maternal health services.

A study from Ghana indicates that compared to non-slum areas, maternal health service coverage is substantially lower in urban slums. Between 2018 and 2021, the attendance of antenatal care, in particular, has dramatically decreased in slum districts as compared to non-slum districts [10].

The sustainable development goal (SDG) emphasized health target 3, aiming to ensure healthy lives and well-being for all. Achieving this requires addressing social determinants of health, particularly in rapidly growing urban areas of low and middle-income countries(LMICs), where inequities, slum proliferation and limited resource disproportionately affect mothers and children [11]. To lessen the significant and growing inequalities in urban areas, which are caused by an increasing number of urban poor people in Ethiopia, improved access to basic healthcare services is required. The Federal Ministry of Health (FMOH) developed the Urban Health Extension Program (UHEP) in 2009 to address the distinct and changing health issues the urban populations face [12].

Several studies on interventions for maternal and newborn babies have demonstrated that the provision of adequate ANC brings with it a positive impact on pregnancy outcomes, through early diagnosis and appropriate management. About 39% of neonatal mortality is reduced due to the provision of adequate ANC to women in Sub-Saharan Africa [13–16].

The World Health Organization in its earlier guideline recommended that every pregnant woman with a normal pregnancy in middle and low-income countries attend at least four ANC visits, and the revised new guideline for safe pregnancy outcomes recommended women to attend eight or more ANC visits [5]. However, many women in low-income countries, particularly in SSA, have limited access and low maternal health service utilization [17]. Several studies have attempted to identify the factors associated with the low utilization of ANC4+ [16,18–21]. Age [16,18–22], Women's education, husbands' education, socioeconomic status [13,15,16,18–22], Women's employment, birth order, media exposure [13,15,16,18–23], and place of residence [19,22,23] are among the factors associated with underutilization of ANC. The factors attributed to less than the recommended number of ANC utilization are quite substantial across regions and sub-populations [18,24].

There is indeed a paucity of information on trends and disparities in antenatal care (ANC) utilization in Ethiopia when disaggregated by urban settings, particularly in relation to both the conventional ANC4+ and the newly recommended ANC8+ models. Most existing studies either focus on national averages or emphasize rural contexts, leaving urban-specific evidence limited despite rapid urbanization and widening inequities in service access. To address this gap, this study aimed to examine trends, disparities, and associated factors of ANC4+ and ANC8+ utilization in urban areas of Ethiopia, using the EDHS series from 2000 to 2019.

## Materials and methods

### Study area and context

This study focused on the urban areas of Ethiopia, using data from the Ethiopian Demographic and Health Survey(EDHS) conducted between 2000 and 2019. Ethiopia is administratively divided into nine regional states and two city administrations, with urban areas spanning major cities and towns across these regions.

### Study design

This study employed a cross-sectional design using EDHS conducted in 2000, 2005, 2011, 2016, and 2019(Mini DHS).

### Data sources

The study used data from 2000 to 2019 EDHS which was obtained from the DHS Program (www.dhsprogram.com) following the required registration and approval process. The surveys are nationally representative and provides estimates of key health and demographic indicators at national and regional levels. The EDHS employs a two stage

stratified cluster sampling design, first selecting enumeration areas(EAs) within each stratum (urban and rural) using probability proportional to size, followed by systematic random sampling of households within the selected EAs. Eligible participants include women aged 15–49 years and men aged 15–59 years residing in the selected households. For this analysis, the inclusion criteria were women of reproductive age (15–49 years) living in urban areas who had at least one live birth within the five years preceding the survey. Women from rural clusters and those with incomplete or missing information on key study variables were excluded. Individual records (IR files) on women aged 15–49 years were used for this analysis. Details on the EDHS study methods are published by Ethiopia's Central Statistics Agency [25–28].

## Variables and measurements

Receipt of ANC was defined as a woman reporting any facility-based ANC visit during the pregnancy of the most recent live birth. The primary outcome variables were ANC utilization, assessed at two thresholds: The first outcome was whether women had at least four ANC visits as "yes" (≥ 4 visits) or "no" (0–3 visits), reflecting the previous WHO recommendation. The second outcome was whether women had at least eight ANC visits, categorized as "yes" (≥8visits) or "no" (0–7 visits), in line with the updated WHO guideline [5]. It was calculated as the percentage of pregnant women who attended four or more or eight or more ANC visits in a given year of the total number of pregnant women aged 15–49 years surveyed in the same year. Trends of ANC service utilization were stratified by wealth status, education status, and region. Wealth status was computed using the country-specific and urban/rural-adjusted wealth quintiles variable (v190a), designed by the DHS team [29]. Then, based on the median wealth index categorized into urban poor and urban non-poor. Maternal educational status was categorized into three categories: no education, primary education, and secondary and higher education. The sub-national regions include nine regions and two city administrations; no sequence was shown in sub-national region segregation.

## Data processing and analysis overview

We used a four-step process: The first involved data assembly, recoding, and harmonizing the DHS datasets. Second, we used descriptive statistics to summarize the Socio-demographic characteristics of the study sample and the distribution of ANC4+ and ANC8 + utilization. This includes calculating frequencies and percentages for categorical variables. Third, we conducted a bivariate logistic regression analyses to assess the crude association between each independent variable and ANC4+ and ANC8 + utilization. Fourth, we fitted a multilevel logistic regression model using the variables identified from the univariate analysis. Analyses were conducted in STATA 16.1 (StataCorp, 2016. Stata Statistical Software version 16.1, College Station, Texas, USA), adjusting for survey design (svyset with clusters, individual sampling weights, and stratification). Finally, 95% CI was computed to measure statistical significance and p-value of <0.05 was used to declare statistical significance.

**Multilevel regression analysis (MLRA).** The study used multilevel regression analysis methods to account for the hierarchical nature of the DHS data by categorizing the outcome variable into binary response. The hierarchical structure of the data follows an individual's (level 1) who are nested within communities (level 2) S1 Fig. The level 1 model represents the relationships among the individual-level variables and the outcome variable, while the level 2 model examines the influence of community-level factors on the outcome variable. The target population was all women aged 15–49 years in randomly selected households across urban areas in nine regions and two city administratives. All urban women aged 15–49 years considered in the series of EDHS were eligible for this study. A total of 6366 urbanite women participated in the five rounds of the EDHS survey were considered for Antenatal care utilization analysis.

According to the current WHO recommendation, a woman is said to utilize optimal ANC if she has eight or more ANC visits at a health facility, but, it is challenging to achieve the eight visit, while ANC4 remains sub-optimal [5]. Accordingly,

this study examined both ANC4+ and ANC8+ utilization as reported across the survey period. The response variable (Yic), four or more ANC, eight or more ANC utilization, is a binary outcome variable and is categorized as:

$$Yic = \begin{cases} 1, & \text{if a woman has 4 or more ANC visits, eight or more ANC} \\ 0, & \text{if less than 4 or no ANC visits, less than eight or no ANC} \end{cases}$$

The null model is used to decompose the total variation into women-level like (education, wealth status, age, marital status, family size, child birth order, working status) and cluster-level (region, religion, year) variations. In the random intercept model, the intercept term is the only random effect; the group differs concerning the average value of the response variable, and the slope is assumed to be fixed across all the regions. The random coefficient model allows the explanatory variables to have different effects for different regions. The functional forms were described below.

$\text{Logit}(\pi ij) = \beta o + uoj$ ⟶ Null model

$\text{Logit}(\pi ij) = \beta o + \sum_{h=1}^{p} \beta h Xhij + uoj$ ⟶ Random intercept model

$\text{Logit}(\pi ij) = \beta o + \sum_{h=1}^{p} \beta h Xhij + \sum_{h=1}^{p} \beta uhj Xhij + uoj$ ⟶ Random coefficient model

Where: $i = 1, 2, 3,....:n_j$ and $j = 1, 2,...; 11$, $\pi_{ij}$ is the probability of attending four or more ANC for $i^{th}$ women in $j^{th}$ region, $\beta 0$ is the population average of the transformed probabilities, $\beta_h$, $h = 1; 2;...; p$ are fixed slope for p explanatory variables, $X_{hij}$; $h = 1; 2;...; p$ are explanatory variables at women level and regional level; $U_{hj}$; $h = 1; 2;...; p$ are random slope for p explanatory variables and $u_{0j}$ is the random effect. $u_{0j}$ ND$(0; \delta_u^2)$, $\binom{uoj}{uhj}$ MND $(0_{p+1}; \mathcal{E}_{p+1})$. Furthermore, $\beta o + \sum_{h=1}^{p} \beta h Xhij$ are assumed to be fixed, and $\sum_{h=1}^{p} \beta uhj Xhij + uoj$ are assumed to be random. The Inter-class Correlation Coefficient (ICC) under the null model is computed as: ICC $= \frac{\sigma 2\, u}{\sigma 2\, u + \sigma e2}$ where $\sigma_e^2$ is women-level variance, which is assumed to be fixed to $\frac{\pi 2}{3} = 3:29$, and $\sigma_u^2$ is regional-level variance, which is obtained from the analysis. The model-building strategy is exploratory and aims to assess how the inclusion of variables and their interactions impacts model improvement. The model-building process was described in S2 Fig. The parameters of the model were estimated by the maximum likelihood method. Model selection was done using Akaike's information criteria (AIC) and Bayesian information criteria(BIC).

Four models having variables of interest were fitted. Model 1: It is the null model, used to check the variability among the communities without inserting exposure variable. It's the first step in MLRA used to provide evidence of whether the data has justifiable evidence to assess the random effects at the community level. Model 2: It is a multivariate model adjustment for individual-level variables that were significant at the bivariate MLRA level. It was built by a stepwise regression analysis technique, which enables the removal of independent variables that become insignificant during the insertion of every other independent variable into the model. Hence, independent variables that were significant in model 2 (i.e., individual-level variables model adjustment) were considered candidates for the final model. Model 3: It is a multivariate model adjustment for community-level variables that were significant at the bivariate MLRA level. All independent variables that were statistically significant in model 3 (community-level variables model adjustment) were included in the final model as potential candidates.

Model 4: It is a multivariate MLRA model adjustment of the outcome variable against predictor variables, which were statistically significant either at the individual or community level model adjustments. In this model, Variables were categorized as community-level or individual-level factors. Community-level factors were survey year, religion, place of previous residence and region; individual-level factors were wealth status, maternal education, maternal age, birth order, partner's educational level, sex of the household head, marital status, husband/partner's educational level, family size, number of living children, exposure to mass media and working status

### Parameter estimations

The measures of association (fixed-effects) estimate the associations between the likelihood of ANC4+/ANC8+ utilization and various explanatory variables measured using an adjusted Odds Ratio (AOR) with their 95% Confidence Intervals (CIs). The measures of variation (random effects) were reported as the intra-community correlation coefficient (ICC), which is the percentage variance explained by the higher level (community level variables). Change in Community Variance (CV), expresses the change in the community-level variance between the empty model (Model 1) and the subsequent models.

### Multicollinearity

The presence of multicollinearity was checked among independent variables using the Variance Inflation Factor (VIF) at the cut-off point of 10. Factors containing a VIF value of less than 10 indicate the absence of multicollinearity. All the variables included in the analysis had a VIF of less than three, indicating the absence of multicollinearity.

### Fitness of the model

Akaike Information Criterion (AIC) was used to estimate the goodness of fit of the adjusted final model in comparison to the preceding models (individual and community level model adjustments). The AIC value for each subsequent model was compared, and the model with the lowest value was considered to be the best-fit model and identify determinant factors of ANC4+ and ANC8+ service utilization.

### Ethics approval and informed consent

As this study involved secondary analysis of publicly available, de-identified data from the EDHS, the requirement for participant consent was waived by the ethics committee of Addis Ababa University. The original DHS surveys received ethical approval from the Ethiopian Public Health Institute Research Ethics Review Committee, the Institutional Review Board of ICF International, and the U.S. Centers for Disease Control and Prevention (CDC). During each survey, informed verbal consent was obtained from all eligible participants prior to the interviews. For respondents younger than 18 years, parental or guardian consent was obtained in accordance with national and international ethical guidelines. All EDHS datasets are fully anonymized before being made publicly accessible. Permission to use the datasets for the present analysis was obtained from the DHS Program (www.dhsprogram.com) following the required registration and approval procedures.

## Results

### Sociodemographic characteristics of the study participants

The pooled mean age of study participants from all the survey year was 28.56±(0.14), Concerning the age distribution of the study participants from all the survey years, the highest proportions (34.8%) of women were in the age group of 25–29 years followed by (19.2%) aged between 30–34 years. Among the study participants 86.5% of them were married. Concerning the education of the study sample from all the survey years, 34.0% of them attended primary education. Nearly half (46.8%) of husbands/partners attended a secondary or higher education level. Over half of the study participants were Orthodox Christians follower (56.9%). Nearly three-quarter (73.9%) of the households were headed by males, and 20.4% of them were between the ages of 30–34. Approximately 66.3% of the study sample belonged to a low wealth status, and 59.5% of the study sample were not working during the survey period (Table 1).

### Exposure to media outlet

The analysis of media exposure among the study sample from the EDHS 2000–2016 reveals notable patterns in the percentage of newspaper/magazine reading, listening radio, and watching television. Over the years, watching TV show the

**Table 1. Socio-demographic characteristics of the study sample from all the survey years, Ethiopian Demographic Health Survey 2000-2019 (n = 6366).**

| Variables | Categories | Frequency | Percentage |
|---|---|---|---|
| Age of women in a complete year | 15-19 | 287 | 4.2 |
| | 20-24 | 1426 | 21.9 |
| | 25-29 | 2156 | 34.8 |
| | 30-34 | 1282 | 19.2 |
| | 35-39 | 860 | 13.0 |
| | 40-44 | 280 | 5.3 |
| | 45-49 | 75 | 1.6 |
| Marital status | Never married | 139 | 2.0 |
| | Married/Cohabitation | 5455 | 86.5 |
| | Divorced/Separated/Widowed | 772 | 11.5 |
| Respondents' Highest level of education | No education | 1980 | 32.5 |
| | Primary | 2023 | 34.0 |
| | Secondary/higher | 2363 | 33.5 |
| Sex of household head | Male | 4461 | 73.9 |
| | Female | 1905 | 26.1 |
| Age of household head in a complete year | 15-19 | 35 | 0.4 |
| | 24−24 | 387 | 6.1 |
| | 25-29 | 1149 | 19.5 |
| | 30-34 | 1302 | 20.5 |
| | 35-39 | 1218 | 18.4 |
| | 40-44 | 813 | 12.9 |
| | 45-49 | 490 | 8.1 |
| | 50+ | 972 | 14.1 |
| Husband/partner's educational level | No formal education | 1035 | 22.1 |
| | Primary | 1430 | 31.1 |
| | Secondary/higher | 2613 | 46.8 |
| Religion | Orthodox | 3205 | 56.4 |
| | Muslim | 2291 | 24.4 |
| | Protestant | 806 | 18.1 |
| | Others | 64 | 1.1 |
| Wealth status | Urban poor | 3418 | 66.5 |
| | Urban non-poor | 2948 | 33.5 |
| Working status | Not working | 3301 | 59.5 |
| | Working | 2061 | 40.5 |

most significant increase, more than tripling over the 15 years (12.7% to 58.4%). The least used media outlet was reading newspaper/magazine changing from 5.0% to 8.7% over 15 years. On the other hand, listening to the radio slightly remain similar over the 15 years (30.3% to 32.7%. When considering all forms of media outlet, slightly increased over time from 10.1% to 28.5% as shown in S3 Fig.

### Reproductive health characteristics and service utilization

The reproductive health characteristics of the study sample using data from all survey rounds(pooled) of EDHS 2000– 2019 revealed that a significant proportion of respondents, 44.8%, had their first birth between the ages of 15 and 19.

Approximately 40.1% of women initiated their first antenatal care (ANC) visit during the second or third month of pregnancy, while a smaller percentage, 4.6%, sought ANC within the first month. More than half of the respondents, 58.9%, delivered in health facilities; however, a substantial 41.1% of births still occurred at home. Regarding the sex of the children, the distribution between male and female children was relatively balanced, with males constituting 51.0% and females 49.0%. Additionally, over half of the respondents, 55.4%, reported having one or two living children(Table 2).

### Regional disparities and trends in antenatal care utilization

The data reveals key trends in antenatal care (ANC) across urban Ethiopia from 2000 to 2019, highlighting improved ANC4+ (four or more visits) coverage in most regions through 2016, followed by a slight decline to 59% by 2019. Addis Ababa consistently led, peaking at 89% in 2016, while regions like Tigray and Benishangul also showed strong gains. In contrast, Somali and Afar remained low, with Somali dropping to 24% in 2019. ANC8+ utilization remained low throughout, rarely surpassing 30%. Addis Ababa demonstrated the highest early rates (38–42%) but dropped to 10% by 2019. Dire Dawa maintained relatively better coverage, while most other regions stayed below 10%, with national ANC8+ utilization falling from 15% in 2005 to just 4% in 2019 S4 Fig.

### Trends of antenatal care utilization in urban Ethiopia

Four or more antenatal care utilization was increased from 43.1% (95% CI: 38.3–48.5) in (2000), to 54.6% (95% CI: 49.8–59.2) in (2005) with an annual change of 5.3%, but, changed from 54.6% in 2005 to 45.5% in 2011 with an annual change of −2.8%. ANC4+utilization during the last 20 years changed from 43.1% (95% CI: 38.3–48.5) in 2000 to 58.7% (95% CI: 52.9–64.2) in 2019, with an average annual change of 0.82%. Eight or more ANC utilization was changed from 11.4%(95% CI: 8.9, 14.8) in (2000) to 15.4% (95%CI: 12.7–18.5%) in 2005, with annual change of 7.0%, but changed from 15.4% in 2005 to 9.8% in 2011 with annual change of −6.1%. Eight or more ANC utilization during the last 20 years changed from 11.4% (95% CI: 8.9–14.8) in 2000, to 3.6% (95% CI: 2.4–5.4) in 2019, with an average annual change of −0.4% (Table 3)

**Table 2. Reproductive health characteristics of the study sample from all survey round(pooled), EDHS 2000-2019.**

| Variables | Categories | Frequency | Percentage |
|---|---|---|---|
| Age of respondent at first birth | 9-14 | 348 | 6.6 |
| | 15-19 | 2694 | 44.8 |
| | 20-24 | 2317 | 35.6 |
| | 25-29 | 774 | 10.4 |
| | 30+ | 236 | 2.6 |
| Timing of first ANC check up | In the 1st one month | 347 | 4.6 |
| | 2-3months | 2,379 | 40.1 |
| | 4months | 1,220 | 24.3 |
| | 5-9months | 1,314 | 31.1 |
| Place of Delivery | Home | 2,043 | 41.1 |
| | Health facility | 4,326 | 58.9 |
| Sex of child | Male | 3260 | 51.0 |
| | Female | 3109 | 49.0 |
| Number of living children | No child | 117 | 1.9 |
| | 1-2 child | 3783 | 55.4 |
| | 3+child | 2469 | 42.7 |

**Table 3. Trends of ANC4+ and ANC8+utilization in Urban Ethiopia from EDHS 2000−2019.**

| Year | ANC4+utilization at national level(urban) | Annual % changes in ANC4+ | 95%CI | ANC8+utilization at national level(urban) | Annual % changes in ANC8+ | 95%CI |
|------|------|------|------|------|------|------|
| **2000** | 43.1% | – | 38.3%−48.5% | 11.4% | – | 8.9% − 14.8% |
| **2005** | 54.6% | 5.3% | 49.8%−59.2% | 15.4% | 7.0 | 12.7% −18.5% |
| **2011** | 45.5% | −2.8% | 41.2%−49.9% | 9.8% | −6.1 | 7.9% − 12.2% |
| **2016** | 62.7% | 7.4% | 57.9%−67.2% | 10.4% | 1.2 | 8.4% − 12.6% |
| **2019** | 58.7% | −2.1% | 52.9%−64.2% | 3.6% | −21.9 | 2.4% − 5.4% |
| | Average annual change | 0.8 | | | −0.4 | |

As shown in S5 Fig, from 2000 to 2019, ANC4+utilization increased among both urban poor and urban non-poor women, with the urban poor rising from 37.8% to 48.7% (a 28.8% increase), and the urban non-poor from 60.5% to 81.8% (a 35.2% increase). In contrast, ANC8+utilization declined significantly for both groups: among the urban poor, it dropped from 7.8% to just 0.6% (a 92.3% decrease), while among the urban non-poor, it fell from 23.3% to 10.5% (a 54.9% decrease.

Maternal education serves as an important equity stratifier, highlighting both absolute and relative differences in antenatal care (ANC) utilization. Among women with secondary or higher education, the proportion reporting ANC4+visits increased from 60.9% in 2000 to 81.3% in 2019, representing an overall increase of 33.5%. In contrast, ANC8+utilization in this group declined significantly from 18.7% to 6.2% over the same period, a 66.8% decrease. For women with no formal education, ANC4+utilization also showed improvement, rising from 25.1% in 2000 to 43.0% in 2019, an overall increase of 71.3%. However, ANC8+coverage in this group dropped from 3.5% to 1.6%, reflecting a 54.3% decrease S6 Fig.

### Factors Associated with ANC utilization

**Factors Associated with ANC4+utilization.** The factors that are significantly associated with ANC4+utilization from pooled data of the EDHS 2000–2019, were women's age, marital status, pregnant women's education, partner/husband's educational level, wealth status, family size, mass media exposure, and childbirth order (p<0.05). Women aged 30–34 years showed the highest ANC4+utilization with significant differences across age groups (p<0.002). The proportion of ANC4+utilization was higher among married women as compared to unmarried women. The proportion of respondents reporting four or more antenatal utilization increased with higher educational attainment (71.4% for those attended secondary or higher education as compared to 33.1% with no education. Similar patterns were observed for husband/partner education; women whose partners had secondary or higher education reported a higher proportion of ANC4+utilization. Similarly, at least once a week media exposure was associated with an increased in the four or more antenatal care utilization (P-value< 0.000) (Table 4).

In urban settings, poor women had significantly lower ANC4+utilizations (41.9%) than urban non-poor women (74.5%) (p<0.001). Regarding birth order, first mother had the highest ANC4+utilization (57.9%), with significant variations across different birth orders (p<0.000). There were significant regional disparities that existed in ANC4+utilization (P-value< 0.000), with Addis Ababa reporting the highest (83.0%) and the Somali region reporting the lowest (22.0%). There was a noticeable trend over the years, with highest ANC4+utilization in 2016, with significant differences across years at (P-value< 0.000) (Table 4).

**Factors Associated with ANC8+utilization.** The factors that are significantly associated with ANC8+utilization were the age of women, pregnant women's education, partner/husband's educational level, wealth status, mass media exposure, working status, and childbirth order (p<0.001). Women aged 30–34 years showed the highest (11.8%)

**Table 4. Differences in ANC4+ and ANC8+utilization status across different categories of exposure variables and their Chi-square test of association in Urban Ethiopia from pooled data of EDHS 2000 to 2019.**

| Variables | Categories | ANC4+ | | Chi-Square | ANC8+ | | Chi-Square |
|---|---|---|---|---|---|---|---|
| | | No N(%) | Yes N(%) | (p-value) | No N(%) | Yes N(%) | (p-value) |
| **Sex of household head** | Male | 2173(46.2) | 2533(53.8) | 2.5(0.111) | 4271(90.7) | 435(9.3) | 2.5(0.115) |
| | Female | 832(50.1) | 828(49.9) | | 1476 (88.9) | 184(11.1) | |
| **Age of the mother** | 15-19 | 160(59.7) | 109(40.3) | 3.5(0.002) | 257(95.4) | 12(4.6) | 3.6(0.002) |
| | 24−24 | 673(48.2) | 723(51.8) | | 1293(92.6) | 104(7.4) | |
| | 25-29 | 1026(46.4) | 1187(53.6) | | 1966(88.9) | 247(11.1) | |
| | 30-34 | 507(41.5) | 714(58.5) | | 1077(88.2) | 144(11.8) | |
| | 35-39 | 372(44.8) | 457(55.2) | | 750(90.6) | 78(9.4) | |
| | 40-44 | 198(58.9) | 138(41.1) | | 305(90.8) | 31(9.2) | |
| | 45-49 | 68(66.9) | 34(33.1) | | 99(97.3) | 3(2.7) | |
| **Marital status** | Never married | 79(62.3) | 48(37.7) | 16.9(0.000) | 107(84.4) | 20(15.6) | 1.5(0.229) |
| | Married/Cohabitation | 2481(45.0) | 3026(55.0) | | 4976(90.4) | 531(9.6) | |
| | Divorced/Separated/Widowed | 445(60.7) | 287(39.3) | | 664(90.7) | 68(9.3) | |
| **Respondents' level of education** | No education | 1384(66.9) | 686(33.1) | 91.9(0.000) | 2003(96.8) | 67(3.2) | 57.1(0.000) |
| | Primary | 1012(46.7) | 1155(53.3) | | 1964(90.6) | 203(9.4) | |
| | Secondary/higher | 608(28.6) | 1521(71.4) | | 1780(83.6) | 349(16.4) | |
| **Husband/partner's level of educational** | No education | 1445(54.4) | 1212(45.6) | 25.5(0.000) | 2544(95.8) | 113(4.2) | 59.0(0.000) |
| | Primary | 749(50.6) | 731(49.4) | | 1355(91.5) | 125(8.5) | |
| | Secondary/higher | 811(36.4) | 1418(63.6) | | 1848(82.9) | 381(17.1) | |
| **Wealth status** | Urban poor | 2461(58.1) | 1771(41.9) | 205.4(0.000) | 4045(95.6) | 186(4.4) | 169.1(0.000) |
| | Urban non-poor | 544(25.5) | 1590(74.5) | | 1702(79.7) | 433(20.3) | |
| **Family size** | 1-5 | 1738(44.8) | 2145(55.2) | 7.2(0.007) | 3505(90.3) | 377(9.7) | 0.005(0.98) |
| | 6+ | 1267(51.0) | 1216(49.0) | | 2241(90.3) | 243(9.7) | |
| **Exposure to Mass Media** | Not at least once a week | 2690(50.5) | 2636(49.5) | 41.8(0.000) | 4922(92.4) | 404(7.6) | 95.2(0.000) |
| | At least once a week | 315(30.2) | 725(69.8) | | 825(79.3) | 215(20.7) | |
| **Working status** | Currently not working | 2068(47.6) | 2279(52.4) | 0.2(0.627) | 3990(91.8) | 356(8.2) | 18.4(0.000) |
| | Currently working | 937(46.4) | 1082(53.6) | | 1757(87.0) | 263(13.0) | |
| **Birth Order** | 1 | 808(42.1) | 1109(57.9) | 14.3(0.000) | 1707(89.0) | 210(11.0) | 5.4(0.001) |
| | 2-3 | 1129(44.3) | 1420(55.7) | | 2265(88.9) | 284(11.1) | |
| | 4-5 | 489(48.3) | 522(51.7) | | 930(92.1) | 81(7.9) | |
| | 6+ | 579(65.1) | 310(34.9) | | 845(95.0) | 44(4.9) | |
| **Religion** | Orthodox | 1494(41.6) | 2097(58.4) | 11.8(0.000) | 3146(87.6) | 445(12.4) | 14.6(0.000) |
| | Muslim | 836(53.9) | 715(46.1) | | 1456(93.9) | 95(6.1) | |
| | Protestant | 622(53.9) | 532(46.1) | | 1075(93.2) | 78(6.8) | |
| | Others | 52(74.4) | 18(25.6) | | 68(98.4) | 1(1.6) | |
| **Region** | Addis Ababa | 182(17.0) | 884(83.0) | 48.2(0.000) | 733(68.9) | 332(31.1) | 52.1(0.000) |
| | Afar | 50(58.1) | 36(41.9) | | 81(94.8) | 4(5.2) | |
| | Amhara | 725(55.2) | 588(44.8) | | 1253(95.4) | 60(4.6) | |
| | Benishangul Gumuz | 30(46.9) | 34(53.1) | | 62(96.1) | 3(3.9) | |
| | Dire Dawa | 34(35.8) | 62(64.2) | | 83(85.8) | 14(14.2) | |
| | Gambella | 26(53.5) | 23(46.5) | | 48(98.3) | 1(1.7) | |
| | Harari | 19(40.9) | 27(59.1) | | 39(84.6) | 7(15.4) | |
| | Oromia | 998(55.5) | 800(44.5) | | 1706(94.9) | 92(5.1) | |
| | SNNPR | 504(54.1) | 428(45.9) | | 890(95.5) | 42(4.5) | |
| | Somali | 257(78.0) | 73(22.0) | | 322(97.5) | 8(2.5) | |
| | Tigray | 180(30.7) | 406(69.3) | | 529(90.3) | 57(9.7) | |

*(Continued)*

**Table 4.** (Continued)

| Variables | Categories | ANC4+ | | Chi-Square | ANC8+ | | Chi-Square |
|---|---|---|---|---|---|---|---|
| | | No N(%) | Yes N(%) | (p-value) | No N(%) | Yes N(%) | (p-value) |
| **Survey Year** | 2000 | 686(56.2) | 534(43.8) | 11.6(0.000) | 1075(88.1) | 145(11.9) | 12.6(0.000) |
| | 2005 | 388(45.5) | 466(54.5) | | 723(84.6) | 131(15.4) | |
| | 2011 | 872(54.5) | 729(45.5) | | 1444(90.2) | 158(9.8) | |
| | 2016 | 487(37.3) | 819(62.7) | | 1171(89.7) | 135(10.3) | |
| | 2019 | 571(41.3) | 813(58.7) | | 1334(96.4) | 50(3.6) | |

ANC8 + utilization with significant differences across age groups (p<0.002). The proportion of respondents reporting ≥8 antenatal care utilization increased with higher educational attainment. Similar patterns were observed for partner/ husband's education; women whose partner had secondary or higher education reported a higher (16.4%) proportion of ≥8 ANC utilization. Similarly, at least once a week media exposure was associated with an increased (20.7%) in the eight or more antenatal care utilization (P-value< 0.000) (Table 4).

In urban settings, poor women had a significantly lower proportion of ANC8 + utilization (4.4%) than urban non-poor women (20.3%) (p<0.000). working status also mattered, as currently working women showed higher (13.0%) ANC8 + utilization than non-working women (8.2%) (p<0.000). Regarding birth order, first time and two-three times mothers had the highest (11.1%) ANC8 + utilization, with significant variations across different birth orders (p<0.001). There were significant regional disparities that exist in ANC8 + utilization (P-value< 0.000), with Addis Ababa reporting 31.1%. There was a noticeable trend over the years, with decreasing ANC8 + utilization from 2000 to 2019, the highest (15.4) ANC8 + utilization observed in 2005 with significant differences across years (Table 4).

## Multilevel analysis of Antenatal care utilization from pooled data of 2000–2019 EDHS

Multilevel binary logistic regression was carried out to identify factors associated with four plus and eight plus visits of antenatal care utilization (ANC4+ and ANC8+). Of the community-level factors examined, religion and geographic area(region) were statistically significantly associated with ANC4+ and ANC8 + utilization in final model and survey year was also considered as level two variable associated with ANC4+ and ANC8 + utilization. Of the individual-level factors, age of respondent, pregnant women's education, partner/husband's educational level and, wealth status showed statistically significant associations with ANC4+ and ANC8 + service utilization in the final model. On the other hand, size of the family, marital status, birth order of the child and exposure to mass media showed statistically significant association with ANC4 + service utilization.

Women aged 30–34 had over 2 times the odds of attending ANC4+ (OR= 2.22; 95% CI: 1.60–3.08) compared to 15–19 years, with 40–44 group had three times higher odds (AOR=3.3; 95% CI: 1.50–3.58). In contrast, women in age group 45–49 years did not demonstrate statistically significant differences in ANC4 + utilization. With regard to ANC8 + utilization, women aged 30–34 years had 1.2 times higher odds (AOR= 1.77; 95% CI: 1.08–2.90) compared to 15–19 years, with 40–44 age group had 2 times higher odds (AOR= 2.11; 95% CI: 1.12–3.97) of ANC8 + utilization compared to 15–19 years' group.

Women who attained secondary and higher educational levels had 2.6 times higher odds of utilizing ANC4 + service as compared to women who haven't been in formal education (OR=2.62; 95% CI: 2.19–3.14). For ANC8 +, the odds of utilizing ANC8 + were 2 times (AOR= 2.13; 95% CI: 1.63–2.78) higher for those who attended secondary and higher educational level. Similarly, women whose husband/partner attained secondary and higher educational level had 1.8 times(OR= 1.81; 95% CI: 1.50–2.19) and 1.6 times (OR=1.56; 95% CI: 1.20–2.02) higher odds of utilizing ANC4+ and ANC8 + services as compared to women whose husband/partner had no formal education, respectively.

Urban non-poor women had substantially higher odds of both ANC4+ (AOR=1.65; 95% CI: 1.42–1.90) and ANC8+ (AOR= 1.96; 95% CI: 1.60–2.40) compared to their urban poor counterparts. On the other hand, mass media utilization was associated with ANC4 + utilization. Women who exposed to mass media at least once a week had 1.5 times higher odds of utilizing ANC4 + service as compared to women who had not been exposed to media at least once a week (OR=1.52; 95% CI: 1.27–2.82). Furthermore, mothers who had first birth order were 1.55 times (OR= 1.55; 95% CI: 1.16–2.07) more likely to utilize ANC4+ as compared to reference group. Women from smaller families (1–5 members) had 21% higher odds of ANC4 utilization as compared to those from larger families (6 + members) (OR= 1.21; 95% CI: 1.06–1.40). Concerning marital status, married or cohabiting women were two times (OR= 2.0; 95% CI: 1.31–3.07) more likely to utilize ANC4+ as compared to never-married women.

After controlling for other variables, the odds of a woman in the Somali region utilizing four or more antenatal care were reduced by 88% than a woman living in Addis Ababa (OR= 0.12; 95% CI: 0.09–0.17). On the other hand, the odds of a woman in the Gambella region utilizing four or more antenatal care were reduced by a dramatic 96% (AOR= 0.04; 95% CI: 0.02–0.10). Furthermore, Women who identify as Protestant (AOR= 0.72; 95% CI: 0.58–0.90) and those belonging to other religions (AOR= 0.52; 95% CI: 0.29–0.93) had significantly lower odds of attending at least four antenatal care (ANC) visits compared to Orthodox women. Additionally, Muslim women were less likely to complete eight or more ANC visits (AOR= 0.80; 95% CI: 0.65–0.99).

Regarding the survey year, by 2016, the odds of ANC4 + utilization had nearly doubled (AOR=1.97) and peaked to (AOR= 2.73) in 2019, indicating steady progress in ANC4 + uptake, while on the other hand, the odds of ANC8 + utilization were significantly lower in both 2016 (AOR=0.66) and 2019 (AOR=0.55) compared to the year 2000(Table 5). For model selection Akaike Information Criterion(AIC) was used, as presented in (Table 6).

## Discussions

This study was conducted to examine trends and disparities of ANC service utilization in urban Ethiopia over the last 20 years using EDHS data. In this study, ANC4 + utilization increased from 43.1% in 2000 to 58.7% in 2019, which means that coverage increased by 15.6 percentage points over the five survey periods. As compared to Nigeria, in this study, ANC4 utilization showed an increment while it slightly decreased in Urban Nigeria from 77.6% in 2013 to 76.1% in 2018 [30]. The increment in Ethiopia may be partly explained by the implementation of the urban health extension program by the Ministry of Health and partner organizations working on maternal health in the country, through health information campaigns about the importance of antenatal care in reducing the complications related to pregnancy, but the overall level of ANC4 + utilization remain lower than the Nigerian study, this could be as result of period of the two studies.

On the other hand, the findings of this study revealed a significant decrease in the utilization of eight or more antenatal care (ANC8+) visits, from 11.4% in 2000 to 3.6% in 2019, reflecting a 7.8 percentage points reduction over five survey rounds. This trend is particularly concerning in light of global efforts to enhance maternal health. One possible explanation for this decline is the adoption of the WHO's Focused Antenatal Care (FANC) model, which was promoted during the survey period by the World Health Organization (WHO) and adopted in many low- and middle-income countries, including Ethiopia. However, in 2016, WHO revised its guidelines and introduced a new antenatal care model recommending a minimum of eight ANC contacts [5]. Despite the launch of this new model, the current findings suggest that its impact had not yet materialized by the time of the 2019 survey. This may reflect a lag in policy adoption, health system readiness, or community-level awareness of the updated guidelines.

Between 2000 and 2019, notable variations in the use of ANC4+ among women aged 15–49 were noted by geographic region and socioeconomic background. ANC4 + utilization increased from 60.5% in 2000 to 81.8% in 2019, with an average annual change of 1.1% per year among the urban non-poor. It shows a slight change (37.8% to 48.7%) among the urban poor over the twenty years' period with an average annual change of 0.6%. This finding implies that there are urban ANC4 + utilization disparities. Even if the maternal health services are provided for free in the governmental facilities of

**Table 5. Multilevel Logistic regression results of individual and community level factors associated with ANC4+ and ANC8+service utilization among women aged 15- 49 years in urban Ethiopia from pooled 2000-2019 EDHS data by socioeconomic and demographic subgroups, EDHS 2000- to 2019.**

| | | ANC4+ | | | | ANC8 | | | |
|---|---|---|---|---|---|---|---|---|---|
| Variable | CCategories | Null Model | Model I AOR(95% CI) | Model II AOR(95% CI) | Model III AOR(95% CI) | Null Model | Model I AOR(95% CI) | Model II AOR(95% CI) | Model III AOR(95% CI) |
| Individual-level variables | | | | | | | | | |
| Age of women | 15-19 | – | 1 | – | 1 | – | 1 | – | 1 |
| | 20-24 | – | 1.55(1.16–2.07)* | – | 1.43(1.08-1.92)* | – | 1.22(0.77–1.95) | – | 1.17(0.73-1.87) |
| | 25-29 | – | 2.15(1.60–2.88)* | – | 1.79(1.33-2.41)* | – | 1.66(1.05–2.64)* | – | 1.45(0.91-2.31) |
| | 30-34 | – | 2.89(2.10–3.98)* | – | 2.22(1.60-3.08)* | – | 2.22(1.37–3.59)* | – | 1.77(1.08-2.90)* |
| | 35-39 | – | 3.03(2.15–4.27)* | – | 2.18(1.54-3.10)* | – | 2.12(1.27–3.54)* | – | 1.56(0.93-2.64) |
| | 40-44 | | 3.18(2.07–4.87)* | | 3.32(1.50-3.58)* | | 2.79(1.49–5.19)* | | 2.11(1.12-3.97)* |
| | 45-49 | | 2.00(1.06–3.78)* | | 1.35(0.71-2.56) | | 2.08(0.76–5.73) | | 1.62(0.58-4.54) |
| Pregnant Women's Education | No Education | – | 1 | – | 1 | – | 1 | – | |
| | Primary | – | 1.85(1.59–2.15)* | – | 1.57(1.35-1.84)* | – | 1.46(1.13–1.88)* | – | 1.49(1.15-1.93)* |
| | Secondary/Higher | – | 3.02(2.54–3.59)* | – | 2.62(2.19-3.14)* | – | 1.99(1.54–2.58)* | – | 2.13(1.63-2.78)* |
| Husband/partner's education | No Education | – | 1 | | 1 | – | 1 | | 1 |
| | Primary | – | 1.26(1.06-1.48)* | | 1.48(1.23-1.79)* | – | 1.43(1.11-1.83)* | | 1.08(0.82-1.44) |
| | Secondary/Higher | – | 1.33(1.13- 1.56)* | | 1.81(1.50-2.19)* | – | 1.91(1.54- 2.36)* | | 1.56(1.20-2.02)* |
| Birth order | 1 | – | 2.34(1.77–3.10)* | – | 1.55(1.16–2.07)* | – | 2.06(1.38–3.08)* | – | 1.45(0.95–2.21) |
| | 2-3 | – | 1.79(1.40–2.28)* | – | 1.32 (1.03 1.70)* | – | 1.73(1.19–2.51)* | – | 1.39(0.94–2.07) |
| | 4-5 | – | 1.46(1.16–1.84)* | – | 1.25(0.98–1.58) | – | 1.55(1.05–2.29)* | – | 1.43(0.96–2.14) |
| | 6+ | – | 1 | – | 1 | – | 1 | – | 1 |
| Family size | 1-5 | | 1.27(1.10–1.45)* | | 1.21(1.06 –.40)* | | NI | | |
| | 6+ | | 1 | | 1 | | | | |
| Marital status | Never married | – | 1 | – | 1 | | NI | | |
| | Married/Cohabitation | – | 1.96(1.30–2.96)* | – | 2.00(1.31-3.07)* | | | | |
| | Divorced/Separated/ Widowed | – | 1.26(0.82–1.93) | – | 1.28(0.82-1.98) | | | | |
| Wealth status | Urban poor | – | 1 | – | 1 | – | 1 | – | 1 |
| | Urban non-poor | – | 2.43(2.12–2.78)* | – | 1.65(1.42–1.90)* | – | 3.02(2.49–3.65)* | – | 1.96(1.60–2.40)* |
| Working status | Currently not working | | NI | | | | 1 | | 1 |
| | Currently working | | | | | | 1.05(.89-1.24) | | 1.06(0.89-1.25) |
| Media exposure | Not at least once a week | – | 1 | – | 1 | – | 1 | – | 1 |
| | At least once a week | – | 1.53(1.28–1.83)* | – | 1.52(1.27 –.82)* | – | 1.31(1.09–1.58)* | – | 1.12(0.93 −1.35) |
| Community level variables | | | | | | | | | |
| Religion | Orthodox | 1 | | 1 | 1 | 1 | | 1 | 1 |
| | Muslim | – | – | 0.57(0.49-0.66)* | 0.87(0.74-1.02) | – | – | 0.54(0.44-0.66)* | 0.80(0.65-0.99)* |
| | Protestant | – | – | 0.73(0.60-0.91)* | 0.72(0.58-0.90)* | – | – | 1.12(0.85-1.47) | 1.08(0.82-1.44) |
| | Others | – | – | 0.58(0.34-1.01) | 0.52(0.29-0.93)* | – | – | 0.47(0.16-1.42) | 0.38(0.12-1.18) |
| Region | Addis Ababa | – | – | 1 | 1 | – | – | 1 | 1 |
| | Afar | – | – | 0.17(0.12-0.24)* | 0.25(0.18–0.35)* | – | – | 0.19(0.11-0.32)* | 0.30(0.17 - 0.51)* |
| | Amhara | – | – | 0.17(0.13-0.23)* | 0.27(0.20–0.36)* | – | – | 0.11(0.06-0.18)* | 0.17(0.10 - 0.29)* |
| | Ben Gumz | – | – | 0.22(0.15-0.32)* | 0.39(0.27–0.56)* | – | – | 0.09(0.04-0.18)* | 0.17(0.08 - 0.35)* |
| | Dire Dawa | – | – | 0.43(0.35-0.54)* | 0.51(0.40–0.64)* | – | – | 0.41(0.32-0.53)* | 0.50(0.39- 0.64)* |
| | Gambella | – | – | 0.20(0.15-0.27)* | 0.31(0.23–0.42)* | – | – | 0.03(0.01-0.06)* | 0.04(0.02 - 0.10)* |
| | Harari | – | – | 0.34(0.27-0.43)* | 0.31(0.24–0.39)* | – | – | 0.46(0.35-0.59)* | 0.43(0.33- 0.56)* |
| | Oromia | – | – | 0.18(0.13-0.23)* | 0.24(0.18–0.32)* | – | – | 0.12(0.07-0.18)* | 0.17(0.11 −0.28)* |
| | SNNPR | – | – | 0.20(0.15-0.28)* | 0.31(0.22–0.43)* | – | – | 0.12(0.07-0.20)* | 0.21(0.12 −0.35)* |
| | Somali | – | – | 0.06(0.04-0.08)* | 0.12(0.09–0.17)* | – | – | 0.08(0.04-0.14)* | 0.14(0.08- 0.27)* |
| | Tigray | – | – | 0.43(0.33-0.58)* | 0.65(0.49–0.87)* | – | – | 0.22(0.16-0.32)* | 0.31(0.22-0.45)* |

*(Continued)*

**Table 5.** (Continued)

| Variable | CCategories | | Null Model | Model I AOR(95% CI) | Model II AOR(95% CI) | Model III AOR(95% CI) | Null Model | Model I AOR(95% CI) | Model II AOR(95% CI) | Model III AOR(95% CI) |
|---|---|---|---|---|---|---|---|---|---|---|
| | | | | ANC4+ | | | ANC8 | | | |
| Survey Year | 2000 | – | – | 1 | 1 | – | – | 1 | 1 |
| | 2005 | – | – | 1.62(1.32-2.00)* | 1.38(1.13-1.70)* | – | – | 1.26(0.98- 1.63) | 1.03(0.79-1.34) |
| | 2011 | – | – | 1.54(1.27-1.87)* | 1.25(1.03-1.52)* | – | – | 1.30(1.02- 1.66)* | 1.14(0.88-1.47) |
| | 2016 | – | – | 2.54(2.09-3.09)* | 1.97(1.62–2.41)* | – | – | 0.86(0.67- 1.10) | 0.66(0.50–0.85)* |
| | 2019 | – | – | 2.15(1.72-2.70)* | 2.73(2.11-3.54)* | – | – | 0.52(0.38-0.72)* | 0.55(0.37- 0.82)* |
| | Intercept | 1.75 (1.59–1.94)* | 0.06 (0.04–0.10)* | 3.70(3.07-4.47)* | 0.18(0.10–0.32)* | 0.14 (0.12–0.16)* | 0.01 (0.01–0.02)* | 0.05(0.41-0.62)* | 0.07(0.03–0.13)* |

Note **: indicates significance level at P-value<0.05, NI: Variable not included the model.

**Table 6. Model Diagnostics and Fit Indices of ANC4+ and ANC8 + Service utilization among women aged 15-49 years in urban Ethiopia from pooled EDHS 2000 to 2019.**

| Model fitness summary for ANC4 + analysis | | | | | Model fitness summary for ANC8 + analysis | | | |
|---|---|---|---|---|---|---|---|---|
| Model Evaluation Metrics | Null Model | Model I | Model II | Model III | Null Model | Model I | Model II | Model III Final |
| Random effect result | | | | | | | | |
| ICC | 18.30% | 8.60% | 6.80% | 3.30% | 24.80% | 13.60% | 7.00% | 5.80% |
| Model fit statistics | | | | | | | | |
| Log-likelihood | −4093.3 | −3602.8 | −3721.9 | −3418.9 | −2547.3 | −2310.5 | −2276.6 | −2167.1 |
| AIC | 8190.5 | 7244.5 | 7481.9 | 6911.9 | 5098..6 | 4657 | 4591.1 | 4404.1 |
| BIC | 8204.1 | 7379.7 | 7610.3 | 7161.9 | 5112.1 | 4778.7 | 4719.6 | 4640.7 |

Ethiopia, women are not equally utilizing the services. Several reasons may explain this gap. For instance, poor people spend more time on activities that generate income needed for daily living than on accessing health services [31]. On the other hand, the time spent waiting in a health facility and the indirect costs related to travel to the health facility discourage poor people from seeking antenatal care. Similarly, significant associations between antenatal care and household wealth status were reported by other studies in Ethiopia [10,19]. Our finding is also similar to other studies reporting that higher incomes are associated with better health-seeking practice and good health status [16]. On the other hand, ANC8 + utilization showed a substantial decline among both urban non-poor and urban poor women between 2000 and 2019. Among the urban non-poor, utilization dropped from 23% to 10.5%, representing a 54.9% decrease. Among the urban poor, it declined even more sharply from 7.8% to 0.6%, marking a 92.3% reduction. These findings highlight growing disparities in antenatal care utilization with urban population, with urban poor experiencing the steepest decline. In this study, maternal education was significantly associated with utilization of both ANC4+ and ANC8 + services. About 71.4% of women who attended at least four ANC visits and 16.4% of those who completed eight or more visits had attained a secondary or higher level of education. This finding is consistent with those of several other studies [16–19,22,24,30]. This finding suggests that education can promote individual health-seeking behavior and, ultimately, increased utilization. The existing evidence indicates that educated mothers may be more likely to be informed about the benefits of antenatal care utilization.

Although many studies have emphasized the advantages of urban residence in accessing and utilizing maternal health services [16,21,23,30]; the present study reveals that urban living alone does not ensure equitable utilization of antenatal

care (ANC) services. The findings show a significant disparities based on wealth status. Specifically, 74.5% of urban non-poor women accessed at least four ANC visits (ANC4+), compared to only 41.6% of urban poor women. The gap is even wider for eight or more ANC visits (ANC8+), with 20.3% of urban non-poor women meeting this threshold, while only 4.4% of urban poor women did the same. These findings highlight that, despite geographic proximity to healthcare facilities, socioeconomic inequality within urban settings remains a major barrier to accessing essential maternal health services. Wealth status clearly emerges as a stronger determinant of ANC utilization than urban residence alone. This underscores the need for targeted interventions within urban areas that address the specific challenges faced by economically disadvantaged women. In this study, no difference was observed between currently working and not working women in ANC4+utilization. This may indicate that ANC4+utilization is more influenced by the economic status of the women than their current working status in the urban context. This finding is consistent with another study from Nigeria [30]. In contrast, a study conducted in Delta State, Nigeria, found that employed women had significantly higher odds of utilizing antenatal care services compared to their unemployed counterparts [32].

There was a marked disparity in antenatal care (ANC) utilization across subnational regions. Addis Ababa had the highest coverage, with 83.0% of women receiving ANC4+ and 31.1% receiving ANC8+, far surpassing other regions. Relatively higher utilization was also observed in Tigray (69.4% ANC4+, 9.7% ANC8+), Dire Dawa (64.2%, 14.2%), and Harari (59.1%, 15.4%), though none showed a statistically significant difference compared to Addis Ababa. Still, Addis Ababa had a 19.8 percentage-point advantage over Tigray, and a 73.5% disparity was observed between the highest and lowest performing regions. At the lower end, Somali region reported only 22.0% utilization, while Gambella recorded only 1.7% ANC8+utilization, highlighting significant regional disparities. These patterns reflect differences in infrastructure, health service availability, and socio-cultural barriers influencing maternal health-seeking behavior, particularly in underserved regions such as the Somali, Afar, and Gambella regions. Similar intra-country disparities have been observed in Nigeria and the Democratic Republic of Congo [33,34], likely reflecting similar socio-economic challenges and regional inequalities in health service delivery. Multilevel binary logistic regression showed that the age of the pregnant women, educational status of the pregnant women, child birth order, family size, marital status, wealth status, religion and region revealed statistically significant associations with four plus antenatal care utilization. For instance, pregnant women whose age lies between 30–35 years were two times more likely to utilize) than mothers who were aged 15–19 years. This is also similar for those aged between 35–39 years. The significant association between maternal age and ANC4+utilization in this study echoes findings from urban settings in Nigeria and Kenya, where older women are more likely to seek comprehensive antenatal care due to increased health awareness and perceived benefit of frequent ANC attendance [20,35]. However, the plateauing effect among women aged 45–49 years may reflect declining perceived necessity or health system engagement in later reproductive years.

The other factor that is significantly associated with ANC4+utilization is education; mothers who attended secondary and higher levels of education were nearly three times more likely to utilize ANC4+ than mothers with no formal education. This finding is consistent with studies from Tanzania and Uganda, where higher education attainment among women directly correlated with increased health service uptake [36,37]. The higher odds of ANC4+utilization among women with secondary or higher education suggest that education not only empowers women to make informed health choices but may also facilitate better utilization of health care and increased health literacy.

The wealth status of a woman is also significantly associated with ANC4+utilization. Urban non-poor women were almost twice more likely to utilize AN4+ than urban poor women. The positive association between urban non-poor and ANC4+utilization is in line with the study conducted in Ghana and South Africa, where economic empowerment translated into greater access to health services [16,38]. In contrast, urban poor women face economic barriers despite geographic proximity to health facilities, underscoring the importance of addressing indirect costs like transportation and opportunity costs related to healthcare access. On the other hand, women from the urban poor often engage in various income-generating activities to cope with the high cost of living in urban areas. The need to earn a livelihood can severely limit their availability and ability to

access maternal health services, including antenatal care. As a result, poor women may prioritize immediate financial needs over their health, leading to inadequate use of antenatal care. Media exposure is significantly associated with ANC4 + utilization. Women who were exposed to media at least once per week were 1.52 times more likely to utilize ANC4+ than their counterparts. This finding aligns with findings from Malawi and Rwanda, where regular media exposure has been shown to increase health knowledge and promote positive health-seeking behavior [21,39]. This highlights the importance of health communication strategies in urban settings to bridge knowledge gaps and encourage ANC4 + utilization.

Other factors significantly associated with ANC4 + utilization were marital status, childbirth order, and family size. Marital status played an important role, married women were found to be two times more likely to utilize AN4 + compared with those who were never married. This finding is also supported by the study from Kenya [20], which suggests that marital support may enhance access and the likelihood of seeking maternal health services. Women experiencing their first birth were 1.55 times more likely to engage in ANC4 + utilization compared to those having six or more children. This could indicate that first-time mothers may be more motivated to seek medical support during pregnancy, possibly due to a lack of prior experience [16]. On the other hand, women with a smaller family size were 1.21 times more likely to utilize ANC4+ than those with six or more family size. This could indicate that large family sizes might create resource constraints or reduce the perceived importance of extensive prenatal care [38]. Overall, these factors highlight the complex interplay between social and family dynamics in shaping maternal health service utilization, particularly the utilization of ANC4 + .

The analysis also revealed several key determinants associated with the utilization of eight or more antenatal care (ANC8+) visits. Maternal age was positively correlated with increased ANC8 + utilization. Women aged 30–34 years and 40–44 years had nearly twice odds of ANC4 + utilization compared to the reference group of 15–19 years. This trend aligns with findings from other studies [32] indicating that older women are more likely to seek comprehensive antenatal care. On the other hand, Maternal education was a significant predictor of ANC8 + utilization. Women with higher educational attainment were more likely to attend eight or more ANC visits. This is consistent with existing literature that underscores the role of maternal education in enhancing health-seeking behaviors [20,40]. In line with this wealth Status played a crucial role in determining ANC8 + utilization. Women from urban non-poor had higher odds of attending eight or more ANC visits, reflecting the influence of financial resources on access to healthcare services [38]

Women who follow Protestant, Muslim, or other religions exhibited reduced odds of utilizing ANC4 + , with Muslim women showing particularly lower odds of attending ANC8 + visits. This may be linked to cultural or religious norms that prioritize same-sex healthcare providers. These preferences, combined with potential gender imbalances in the healthcare workforce, could contribute to lower utilization of ANC services within the religious communities. These are also supported by different studies [41,42].

Generally, this study highlights the multifaceted nature of ANC utilization in urban Ethiopia, influenced by individual, economic, and regional factors. Addressing these disparities requires tailored interventions that enhance health education, reduce economic barriers, and improve service delivery, particularly in underserved regions. Future research should explore context-specific strategies to enhance equitable access to maternal health services across diverse urban settings.

## Strengths and limitations

The Ethiopian Demographic Health Survey is a nationally representative household survey implemented regularly and provides health information nationally, and data is analyzed using a multilevel model to identify the individual and community level factors. However, EDHS data collections were subject to several sources of bias. One of them is reliance on self-reports of ANC service utilization, with the possibility of recall bias.

## Conclusions

This study showed a positive trend in the utilization of four or more antenatal care (ANC4+) visits among urban women, increasing from 43.1% in 2000 to 58.7% in 2019. However, the utilization of eight or more ANC visits (ANC8+) declined

significantly, from 11.4% in 2000 to 3.6% in 2019. These trends underscore persistent disparities in ANC utilization based on wealth status, educational level, marital status, media exposure, and geographic location. To improve ANC4+ and ANC8+utilization, particularly among urban poor women, targeted intervention, media awareness creation, and educational initiatives should be prioritized. Additionally, addressing economic barriers and fostering collaboration among stakeholders will be crucial in promoting equitable access to antenatal care services.

## Supporting information

**S1 Fig. Hierarchical data structure.**
(TIF)

**S2 Fig. Model-building process for a two-level hierarchical model.**
(TIF)

**S3 Fig. Trends in exposure to different forms of media outlet among women in urban Ethiopia, EDHS 2000–2016.**
(TIF)

**S4 Fig. Regional Disparities and Trends in ANC4+ and ANC8+Antenatal Care Utilization in Urban Ethiopia, from EDHS 2000–2019.**
(TIF)

**S5 Fig. Trends of ANC4+ and ANC8+service utilization by years and wealth status in urban Ethiopia from EDHS 2000–2019.**
(TIF)

**S6 Fig. Trends of ANC4+ and ANC8+service utilization by years and educational level in urban Ethiopia from EDHS 2000–2019.**
(TIF)

**S1 File. Dataset used for ANC4_ANC8 Analysis.**
(DTA)

## Acknowledgments

The authors would like to thank major DHS center for sharing the data used for the analysis for free. We would also acknowledge Addis Ababa University, school of Public Health and individuals who were directly or indirectly participated in the preparation of this manuscript.

## Author contributions

**Conceptualization:** Worku Dechassa Heyi, Wubegzier Mekonnen Ayele, Dessalegn Y. Melesse.

**Data curation:** Worku Dechassa Heyi, Wubegzier Mekonnen Ayele, Dessalegn Y. Melesse.

**Formal analysis:** Worku Dechassa Heyi, Wubegzier Mekonnen Ayele, Dessalegn Y. Melesse.

**Investigation:** Worku Dechassa Heyi, Wubegzier Mekonnen Ayele.

**Methodology:** Worku Dechassa Heyi, Wubegzier Mekonnen Ayele, Dessalegn Y. Melesse.

**Project administration:** Worku Dechassa Heyi.

**Software:** Worku Dechassa Heyi, Wubegzier Mekonnen Ayele, Dessalegn Y. Melesse.

**Supervision:** Wubegzier Mekonnen Ayele, Dessalegn Y. Melesse.

**Validation:** Worku Dechassa Heyi, Wubegzier Mekonnen Ayele.

**Visualization:** Worku Dechassa Heyi, Dessalegn Y. Melesse.

**Writing – original draft:** Worku Dechassa Heyi, Wubegzier Mekonnen Ayele, Dessalegn Y. Melesse.

**Writing – review & editing:** Worku Dechassa Heyi, Wubegzier Mekonnen Ayele, Dessalegn Y. Melesse.

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
