## [Decision Letter · Decision Letter 0]

8 Sep 2025

Dear Dr. Heyi,

Thank you for submitting your manuscript to PLOS ONE. After careful consideration, we feel that it has merit but does not fully meet PLOS ONE’s publication criteria as it currently stands. Therefore, we invite you to submit a revised version of the manuscript that addresses the points raised during the review process.

We look forward to receiving your revised manuscript.

Kind regards,

Dubale Dulla Koboto, MSc

Academic Editor

PLOS ONE

**Journal Requirements:**

1. When submitting your revision, we need you to address these additional requirements. Please ensure that your manuscript meets PLOS ONE's style requirements, including those for file naming. The PLOS ONE style templates can be found at https://journals.plos.org/plosone/s/file?id=wjVg/PLOSOne_formatting_sample_main_body.pdf and https://journals.plos.org/plosone/s/file?id=ba62/PLOSOne_formatting_sample_title_authors_affiliations.pdf 2. Thank you for uploading your study's underlying data set. Unfortunately, the repository you have noted in your Data Availability statement does not qualify as an acceptable data repository according to PLOS's standards. At this time, please upload the minimal data set necessary to replicate your study's findings to a stable, public repository (such as figshare or Dryad) and provide us with the relevant URLs, DOIs, or accession numbers that may be used to access these data. For a list of recommended repositories and additional information on PLOS standards for data deposition, please see https://journals.plos.org/plosone/s/recommended-repositories. 3. Please amend either the abstract on the online submission form (via Edit Submission) or the abstract in the manuscript so that they are identical. 4. Your ethics statement should only appear in the Methods section of your manuscript. If your ethics statement is written in any section besides the Methods, please move it to the Methods section and delete it from any other section. Please ensure that your ethics statement is included in your manuscript, as the ethics statement entered into the online submission form will not be published alongside your manuscript. 5. We notice that your supplementary figures are uploaded with the file type 'Figure'. Please amend the file type to 'Supporting Information'. Please ensure that each Supporting Information file has a legend listed in the manuscript after the references list. 6. If the reviewer comments include a recommendation to cite specific previously published works, please review and evaluate these publications to determine whether they are relevant and should be cited. There is no requirement to cite these works unless the editor has indicated otherwise. 

Reviewers' comments:

**Comments to the Author**

1. Is the manuscript technically sound, and do the data support the conclusions?

Reviewer #1: Yes

2. Has the statistical analysis been performed appropriately and rigorously?

Reviewer #1: Yes

3. Have the authors made all data underlying the findings in their manuscript fully available?

Reviewer #1: Yes

4. Is the manuscript presented in an intelligible fashion and written in standard English?

Reviewer #1: Yes

**Reviewer #1:**  Reviewer Comments to author by section

Corresponding name

1. Line 3, the corresponding author name is written as Work, check if it may spelled in wrong way to mean Worku.

Abstract

2. Line 35, the author used 96% CI with 5% level of significance. Why 96% CI with 5% alpha?

Variables and measurements

3. Line 147, the author categorized the outcome variable into four or more or eight or more ANC visits. What is the baseline and purpose of these categories?

4. Grouping as the number of ANC as "4+" and "8+" causes double-counting (e.g., 9 visits belongs to both categories). This may make the parameter estimation over fit and challenge to compare both groups. It is better to separate the overlapping of outcome categories like 4-7 ANC and 8+ or mutually exclusive categories.

Results

5. Perhaps, attending ANC ≥8 visits is not clinically harmful and often reflects strong engagement with prenatal care. However, authors investigated factors related to attending ANC visits 8 and more. Why study factors behind ANC ≥8 visits?

6. Why factors of both ANC 4+ and 8+ are differ?

**Do you want your identity to be public for this peer review?** For information about this choice, including consent withdrawal, please see our Privacy Policy

Reviewer #1: **Yes: ** Senahara Korsa Wake

---

## [Author Response · Author response to Decision Letter 1]

24 Sep 2025

Response to Reviewers

1. What will be the outcome/impact of this study for your community?? Because, there was variations in application of guidelines among conventional ANC, ANC4 and newly introduced ANC contact?

• Thank you for this valuable question. The main outcome is to provide evidence on trends and determinants of ANC utilization under the conventional and ANC4 guidelines, which remain highly relevant for understanding service coverage and equity gaps. Our findings also demonstrate variations in the application of ANC guidelines across time and population groups, underscoring persistent disparities in access and utilization. While the newer ANC8 model was introduced later, these results provide important baseline patterns and barriers that can inform future program adjustments and help policymakers anticipate challenges in scaling up the new guideline.

2. Check Work or Worku?

• We apologize for the earlier misspelling and thank you for bringing it to our attention. The name has been corrected to Worku

3. Does this statement describe the intention of the study???

• Thank you for this important observation. Yes, the statement reflects the intention of the study, which was to examine disparities and trends in antenatal care utilization among urban women in Ethiopia. Specifically, the study assessed both four or more ANC visits (ANC4+) and eight or more ANC visits (ANC8+) between 2000 and 2019, and identified associated factors influencing utilization. Additionally, we have revised the objective for clarity in the manuscript as follow: “This study aimed to examine disparities, trends, and associated factors of antenatal care utilization (ANC4+ and ANC8+) among urban women in Ethiopia between 2000 and 2019.”

4. Anyone, sure that ANC contact (ANC 8) was implemented during the underlined survey??

• Thank you for raising this important point. We acknowledge that the ANC8+ model was introduced by WHO in 2016, after most surveys rounds. The EDHS up to 2019 collected the number of ANC visits but did not explicitly reference ANC8+. Thus, while ANC8+ was not formally implemented during the earlier surveys, the data still allows us to classify women by the number of ANC visits received, including those who had eight or more visits. Our intention in analyzing both ANC4+ and ANC8+ was to assess utilization trends against the conventional benchmark (ANC4+) and to provide baseline evidence on higher-level service uptake (ANC8+), which is relevant for anticipating the feasibility and challenges of adopting the new WHO guideline in Ethiopia.

5. unnecessary, since the data indicates it independently.

• Thank you for this comment. The statement is revised to avoid unnecessary detail.

6. This is an unexpected literal definition in a such manner.

• We thank the reviewer for commenting this part. Now the definition is revised and paraphrased to meet the literature definition.

7. A long paragraph under a single citation.

• Thank you very much for the constructive comments. Now the paragraph is paraphrased and shortened.

8. too huge paragraph under a single citation. Line 104

• Thank you again for reflecting this comment. The paragraph is paraphrased and shortened by removing unnecessary detail.

9. I can't see the statement problem that indicate the shortage of information under the particular study title and even in the country. This is to mean the statement of problem in insufficient.

• We appreciate this insightful comment. We agree that a clear statement of the problem is essential. Now we have clearly stated the statement of the problem in the manuscript.

10. State the study design clearly, here.

• Thanks for the constructive comments. The study design is added to the methods section and the paragraph is also revised to make it clear and coherent.

11. Why rural community was excluded?? Any pointing out the inclusion and exclusion criteria?

• Thank you for this important comment. Our study specifically focused on the urban population, as the primary objective was to examine maternal healthcare utilization within rapidly growing urban settings where unique challenges, such as population density, inequitable access to services, and urban poverty differ significantly from rural contexts. Rural communities were therefore excluded to maintain a clear analytic focus and to avoid masking urban-specific patterns with broader national averages.

12. How can you reconcile the above highlighted clauses?

• We thank the reviewer for pointing out this valid concern. However, the outcome variable was revised to avoid confusion and it is important to note that the ANC4 and ANC8 outcomes are not mutually exclusive. Women with eight or more ANC visits were classified as “yes” for both ANC4 (≥4 visits) and ANC8 (≥8 visits). Women with four to seven visits were classified as “yes” for ANC4 but “no” for ANC8, while those with fewer than four visits were classified as “no” for both outcomes. This approach allowed us to assess ANC utilization against both the earlier WHO recommendation of at least four visits and the updated guideline of at least eight contacts.

Line 35, the author used 96% CI with 5% level of significance. Why 96% CI with 5% alpha?

• Thank you for catching this error. The use of “96% CI” in line 35 was an editorial mistake. We consistently used a 95% confidence interval with a 5% level of significance throughout the analysis, and the manuscript has been corrected accordingly. Thanks,

Line 147, the author categorized the outcome variable into four or more or eight or more ANC visits. What is the baseline and purpose of these categories?

• Thank you for this important question. The outcome variable was categorized into ≥4 ANC visits (ANC4+) and ≥8 ANC visits (ANC8+) based on WHO recommendations. The ANC4+ threshold reflects the earlier WHO guideline, which remained the standard during most of the survey years analyzed. The ANC8+ threshold corresponds to the updated 2016 WHO guideline, which, although introduced later, provides a useful benchmark to explore higher levels of service uptake. Using both categories allowed us to assess trends against the conventional standard while also establishing baseline evidence for the newer ANC8+ model.

Grouping as the number of ANC as "4+" and "8+" causes double-counting (e.g., 9 visits belongs to both categories). This may make the parameter estimation over fit and challenge to compare both groups. It is better to separate the overlapping of outcome categories like 4-7 ANC and 8+ or mutually exclusive categories.

• Thank you for raising this thoughtful point. We agree that ANC4+ and ANC8+ are not mutually exclusive categories, as women with ≥8 visits are also included in the ≥4 group. Our intention in using both thresholds was not to compare them directly but to reflect two distinct global benchmarks: the former WHO guideline (≥4 visits) and the updated 2016 guideline (≥8 visits). To avoid misinterpretation, we clarified in the manuscript that ANC4+ and ANC8+ analyses were conducted independently, each serving as a separate indicator of service uptake rather than mutually exclusive categories. This approach provides complementary insights into utilization under both standards without implying overlap-based comparisons.

Perhaps, attending ANC ≥8 visits is not clinically harmful and often reflects strong engagement with prenatal care. However, authors investigated factors related to attending ANC visits 8 and more. Why study factors behind ANC ≥8 visits?

• We thank the reviewer for this comment. Although attending ≥8 ANC visits is not clinically harmful, it reflects a higher level of engagement with maternal healthcare, which is important for understanding the uptake of the updated WHO guideline. Investigating factors associated with ANC8+ allows us to identify barriers and facilitators for achieving optimal care, which can inform strategies to improve maternal and neonatal outcomes in urban Ethiopia.

Why factors of both ANC 4+ and 8+ are differ?

• The factors of ANC4+ and ANC8+ may differ because ANC8+ represents a more demanding standard of care, requiring more frequent visits and higher continuity. Socioeconomic, geographic, and health system factors that enable women to achieve at least four visits may not be sufficient to reach eight visits. By analyzing both, we capture nuances in utilization patterns and identify specific gaps in achieving the newer WHO-recommended standard.

---

## [Editor Report · Decision Letter 1]

6 Nov 2025

Trends and Disparities in Antenatal Care Utilization: A Multilevel Analysis of Four-plus and Eight-plus ANC Visits among Urban Women in Ethiopia

PONE-D-25-24967R1

Dear Dr. Heyi,

We’re pleased to inform you that your manuscript has been judged scientifically suitable for publication and will be formally accepted for publication once it meets all outstanding technical requirements.

Kind regards,

Dubale Dulla Koboto, MSc

Academic Editor

PLOS ONE
---

## [Editor Report · Acceptance letter]

PONE-D-25-24967R1

PLOS ONE

Dear Dr. Heyi,

I'm pleased to inform you that your manuscript has been deemed suitable for publication in PLOS ONE. Congratulations! Your manuscript is now being handed over to our production team.

Kind regards,

on behalf of

Dr. Dubale Dulla Koboto

Academic Editor

PLOS ONE